# Viability and Desiccation Resistance of *Bartonella henselae* in Biological and Non-Biological Fluids: Evidence for Pathogen Environmental Stability

**DOI:** 10.3390/pathogens12070950

**Published:** 2023-07-18

**Authors:** Janice C. Bush, Ricardo G. Maggi, Edward B. Breitschwerdt

**Affiliations:** Intracellular Pathogens Research Laboratory, Department of Clinical Sciences, Comparative Medicine Institute, College of Veterinary Medicine, North Carolina State University, Raleigh, NC 27606, USA; jcbush@ncsu.edu (J.C.B.); rgmaggi@ncsu.edu (R.G.M.)

**Keywords:** *Bartonella*, vector-borne, zoonotic, intracellular, alphaproteobacteria, desiccation, environment

## Abstract

Pathogen environmental stability is an often-neglected research priority for pathogens that are known to be vector-transmitted. *Bartonella henselae*, the etiologic agent of Cat Scratch Disease, has become a “pathogen of interest” in several serious human illnesses, which include neoplastic, cardiovascular, neurocognitive, and rheumatologic conditions. Survival in the flea gut and feces as well as the association with a biofilm in culture-negative endocarditis provides insight into this organism’s ability to adjust to environmental extremes. The detection of *B. henselae* DNA in blood and tissues from marine mammals also raises questions about environmental stability and modes of pathogen transmission. We investigated the ability of *B. henselae* to survive in fluid matrices chosen to mimic potential environmental sources of infective materials. Feline whole blood, serum and urine, bovine milk, and physiologic saline inoculated with a laboratory strain of *B. henselae* San Antonio 2 were subsequently evaluated by culture and qPCR at specified time intervals. Bacterial viability was also assessed following desiccation and reconstitution of each inoculated fluid matrix. *Bartonella henselae* SA2 was cultured from feline urine up to 24 h after inoculation, and from blood, serum, cow’s milk, and physiologic saline for up to 7 days after inoculation. Of potential medical importance, bacteria were cultured following air-desiccation of all fluid inoculates. The viability and stability of *Bartonella* within biological and non-biological fluids in the environment may represent a previously unrecognized source of infection for animals and human beings.

## 1. Introduction

*Bartonella henselae* is an increasingly important emerging zoonotic vector-borne pathogen, with a worldwide distribution among cats, other mammals, and *Ctenocephalides felis* fleas [1,2,3,4]. Currently, 75% of emerging infectious diseases are considered zoonotic, and 28% of these infections are transmitted by one or more vectors [5,6,7]. To protect human and animal health, it is critical to determine potential exposure risks and infectivity of these bacteria in the environment, in addition to ongoing efforts to elucidate each pathogen’s zoonotic and vector potential. It is increasingly clear that members of the genus *Bartonella*, all of which are proven or suspected to be vector-borne endotheliotropic [8,9] and intra-erythrocytic pathogens [10,11], are responsible for a variety of emergent or re-emergent diseases worldwide, including recent outbreaks of urban “Trench Fever” (*Bartonella quintana*) in Denver, Colorado [12], and bacillary angiomatosis (*Bartonella henselae*, *Bartonella quintana*) in immune-competent patients associated with skin trauma or following solid organ transplantation in immunocompromised patients [13,14]. Of the sixteen *Bartonella* species reported to cause disease in humans, *B. henselae*, the etiologic agent of Cat Scratch Disease (CSD), has become a primary pathogen of interest. Compared to other *Bartonella* species, *B. henselae* has been associated with numerous novel chronic medical conditions, collectively termed “Bartonellosis”, in deference to the acute diagnosis of CSD [15]. The ability of *B. henselae* to infect and persist in a wide range of cell types [10,11,16,17,18,19,20], along with its ability to subvert and modulate the host immune response [21,22], affords the bacteria access to virtually any organ system, a fact underscored by its potential role in a diversity of human pathology [6,13,14,22,23,24,25,26,27,28,29,30,31,32,33,34,35,36,37,38,39,40,41,42,43,44,45], thereby making this zoonotic pathogen species of particular biomedical importance. 

Given the increasing number of diseases associated with *Bartonella* infection, it is important to consider and assess potential environmental risks, as well as sources of human infection other than vector transmission of this organism. Human transmission through subcutaneous inoculation of flea excrement from an animal scratch is the most commonly accepted source of infection [1]. In terms of human exposure, multiple wild and domestic mammals with spatial proximity to humans serve as reservoir hosts for *Bartonella* species [45]. Although the domestic cat (*Felis catus*) is the primary reservoir for *B. henselae*, documentation of these bacteria in other mammalian species, including wild felids [46,47], raccoons [48], domestic dogs [2,3,49], horses [50], cattle [51,52], and feral swine [53], may translate to increased human exposure risks. Research on potential arthropod vector species has shown varied results, as bacterial presence does not denote vector competency in transmitting the bacteria. However, aside from the vector-competent cat flea, documented cases of human infection have been associated with exposure to arthropods, including hard-bodied ticks [54,55,56,57], ants [58], and spiders [59]. Therefore, further investigation of potential vector species, as well as sources of vector infection, need to be addressed [60]. Direct *B. henselae* blood transmission resulting from a needle-stick from an infected cat has been documented [61], bacteremia has been reported in blood donors, and *B. henselae* remains viable in blood transfusion products for weeks [62,63]. Additionally, bite wound transmission from infected animals and the presence of *B. henselae* DNA in the saliva of cats [64] and dogs [65] suggest that transmission may be occurring from exposure to bodily fluids other than blood. Collectively, these findings suggest that direct vascular contact with infected blood or other biological fluids might result in *B. henselae* transmission.

Perhaps most intriguing is the identification of *B. henselae* DNA in blood or tissue samples from various marine mammal species [66,67]. Arguably, species that spend time on land, including sea otters [68] and harbor seals [69], may have exposure to a terrestrial vector species or domestic or wild animal reservoir hosts. However, fully aquatic species such as dolphins, porpoises, and whales lack this potential terrestrial route of exposure, and the mode of transmission of *B. henselae* to these animals has not been determined. First documented in blood from two live-stranded porpoises [70], *B. henselae* has since been identified in several other cetacean species [66,67]. A study evaluating *Bartonella* species DNA from coastal stranded cetaceans compared to blood obtained from pelagic dolphins (*Tursiops truncatus*) found an interesting difference in prevalence. *Bartonella* was identified in 43% of the stranded animals compared to 2.8% of the pelagic animals [66], which posed the question: Is there a coastal geographic risk of *B. henselae* transmission among marine mammals? Other terrestrial pathogens that infect aquatic species, termed “pathogen pollution”, often stem from fecal contamination of watersheds located in proximity to coastal areas. *Toxoplasma gondii* [71] and *Sarcocystis neurona* [72,73] infections in marine mammals are examples of protozoal pathogens linked to exposure to terrestrial animals and their excrement, where freshwater runoff and invertebrate pathogen concentrations allow for infectious stages of these organisms to be ingested by aquatic species. *Leptospira interrogans*, a spirochete bacterium, presents another example of a terrestrial animal pathogen that has acquired a niche among marine species, gaining access through skin lesions or across mucus membranes [74,75,76,77]. Although the ecology of exposure among aquatic animal species remains to be elucidated, *B. henselae* is ubiquitously present worldwide and causes long-lasting relapsing bacteremia in domestic and feral cats [78]; therefore, *B. henselae* may be contaminating the environment of coastal ocean-dwelling animal species. 

Other than a study that documented *B. henselae* survival in flea feces for up to 12 days [1], we are unaware of other research that has addressed the environmental stability of this bacterial species. To further investigate environmental stability, we evaluated the ability of *B. henselae* to survive in various fluids, chosen to mimic fluids from an infected host, which could be a potential source of environmental spillover into terrestrial and aquatic environments. We also questioned whether *B. henselae* could survive desiccation following culture in various fluid matrices. Given that cats are the primary reservoir host and sustain long-standing bacteremia [78], feline whole blood, serum, and urine were chosen as fluid matrices for this study. Bovine milk was also tested, as *B. henselae* bacteremia has been infrequently documented in cows [52]. Physiologic saline was selected to approximate the salinity in the coastal marine environment. Brugge, a liquid mammalian cell culture medium, was chosen as a control fluid [79]. We hypothesized that *B. henselae* would not remain viable in any of the fluid matrices except feline blood, and that the bacteria would not remain viable following desiccation in any fluid matrix. 

## 2. Materials and Methods

### 2.1. Study Design

To address these hypotheses, we first evaluated the ability of *B. henselae* strain San Antonio 2 (*Bh* SA2) to survive in the various fluids for a period of up to 7 days, through direct culture and qPCR amplification of DNA to detect trends in bacterial genome equivalents (GE) over time. To assess the ability to survive desiccation, sequential fluid inoculates were allowed to air-desiccate for seven days, reconstituted in Brugge medium, and then assessed for viability through agar plate culture. The overall study design is depicted in Figure 1. 

### 2.2. Type and Source of Fluid Matrices

Feline whole blood, serum, and urine were obtained commercially (pooled samples from healthy male and female cats) from Biochemed Services (Winchester, VA, USA). Whole, ultra-pasteurized, organic cow’s milk was purchased from a local retail grocery store, and laboratory-grade 0.9% physiologic saline was obtained from Intermountain Lifesciences (Salt Lake City, UT, USA, cat.# Z1376). Brugge medium, dedicated for experimental use only and sterilely prepared in-house, was used as a positive control culture matrix, to provide culture enrichment following inoculation of test fluid matrices, and for reconstitution of fluid matrices following desiccation.

### 2.3. Pre-Inoculation Evaluation of Fluid Matrices for Bartonella Species DNA and Bacterial Growth

Prior to inoculation with *Bh* SA2, 100 µL of each fluid was plated on TSA with 5% sheep blood (Thermo Scientific, Raleigh, NC, USA, cat.# R01200) and incubated at 35 °C/ 5% CO_2_. In addition, paired 250 µL samples were interrogated for the presence of *B. henselae* DNA by qPCR amplification targeting the *Bartonella* 16S-23S intergenic spacer (ITS) region (see Section 2.4). As cats are the known reservoir of *B. henselae*, antibody screening was performed to assess for the presence of anti-*Bartonella* antibodies that could impact bacterial survival in serum. Serum was assessed by immunofluorescence antibody testing (IFA) for the presence of *Bartonella* species-specific antibodies through the Vector Borne Disease Diagnostic Laboratory (VBDDL) at the North Carolina State University College of Veterinary Medicine. All samples were screened for *B. henselae*, *B. koehlerae*, and *B vinsonii* subspecies *berkhoffii* with titer dilutions tested between 1:16 and 1:8192. IFA antigens were grown in vitro by personnel at the VBDDL in DH82 cells (canine macrophage line) used for fluorescent antibody assays. Slides were prepared and assessed in-house using a Zeiss Axio Lab A1 ultraviolet microscope (Fisher Scientific, Waltham, MA, USA, cat.# 12-071-321) under 40× objective. End-point titers ≥1:64 were considered positive in order to account for the potential of a dilution effect subsequent to the use of pooled serum samples from cats with unknown *Bartonella* exposure without overinterpretation. 

### 2.4. Preparation of Bacterial Stock for Inoculation of Fluid Matrices

A low passage (passage # 5) of *Bh* SA2 was grown on TSA with 5% sheep blood (Thermo Scientific, Raleigh, NC, USA, cat.# R01200) incubated at 35 °C/5% CO_2_. Five to six colonies, removed via a sterile culture loop, were inoculated into 10 mL of freshly prepared Brugge culture medium in a T-25 tissue flask and allowed to grow in an incubator at 35 °C/5% CO_2_ for 4 days. The negative control, an un-inoculated flask containing 10 mL of Brugge medium, was prepared simultaneously and incubated alongside the *Bh* SA2-inoculated stock preparation. Manual DNA extraction (Qiagen DNeasy Blood and Tissue Kit, Germantown, MD, USA, cat.# 69504) following the manufacturer’s protocol was performed on paired 250 µL aliquots obtained from the *Bh* SA2 stock and the negative control flasks. Stock bacterial concentration was determined through qPCR amplification of the *Bartonella* 16S-23S ITS region using primers:BsppITS325s: 5′CCTCAGATGATGATCCCAAGCCTTCTGGCG 3′ andBsppITS543as: 5′AATTGGTGGGCCTGGGAGGACTTG 3′.

A BioRad CFX Opus 96 Real-Time PCR System (BioRad, Hercules, CA, USA, cat.# 12011319) was used for all qPCR testing under the following conditions: 95 °C × 5 min for enzyme activation, followed by 45 cycles of 94 °C × 10 s for denaturation, 68 °C × 10 s for annealing, and 72 °C × 15 s for elongation. PowerUp SYBR Green Master Mix (Thermo Fisher, Raleigh, NC, USA, cat.# A25741) was utilized for all experiments in a total reaction volume of 25 µL. GE were plotted against known bacterial DNA stock dilutions to determine the *Bartonella* concentration in the inoculum culture.

### 2.5. Culture, DNA Extraction, and qPCR Evaluation of the Inoculated Fluid Matrices

Fluid matrices were stored at temperatures recommended for stability prior to use; feline blood, cow’s milk, and Brugge medium were refrigerated at 4 °C, feline serum and urine were frozen at −20 °C, and saline was stored at ambient room temperature (20–22 °C). All fluids were brought to ambient temperature before being aliquoted into 10 mL volumes and instilled into sterile T-25 tissue flasks. Each experimental fluid was inoculated with 500 µL of the bacterial stock concentration to result in 10^9^ bacteria per µL of fluid matrix. As the positive control, 10 mL of Brugge medium was inoculated, and the original un-inoculated Brugge flask was maintained as the negative control. Following inoculation, the flasks were gently agitated to allow for bacterial distribution, and time 0 h samples were obtained as follows: paired 250 µL aliquots were placed into 1.8 mL cryovials for storage at −20 °C pending DNA extraction, and 100 µL was plated onto TSA with 5% sheep blood (Figure 1a). Then, 100 µL was placed into 5 mL of fresh Brugge medium, used for bacterial culture enrichment, in a sterile T-25 tissue flask (Figure 1b). Paired 250 µL aliquots were instilled into individual wells of paired 6-well plates for desiccation (Figure 1c). All timed collections followed the above protocol (Figure 1). 

Brugge culture enrichment flasks and inoculated agar plates were placed under incubation at 35 °C/5% CO_2_. Agar plates were evaluated for colony formation every four to seven days, and colonies, removed via a sterile inoculating loop, were placed into 100 µL of Buffer AL (Qiagen, Germantown, MD, USA, cat.# 19075) for subsequent DNA extraction, qPCR, and DNA sequencing. Brugge culture enrichment flasks were sampled at 7, 14, and 21 days post-inoculation: paired 250 µL aliquots were collected for manual DNA extraction and qPCR, and 100 µL was instilled onto TSA with 5% sheep blood (Figure 1b). Subculturing original fluid inoculates into Brugge medium was performed to assess bacterial viability in a known growth medium following incubation in an experimental fluid matrix [79]. As all experiments were run concurrently, the viability of original fluid inoculates was unknown at the onset of the experiment.

### 2.6. Desiccation of the Inoculated and Un-Inoculated Fluid Matrices

Sterile lidless 6-well plates were placed into a biosecurity cabinet with positive airflow overnight to allow for fluid desiccation prior to being fitted with lids and placed into an enclosed benchtop container maintained at ambient temperature for seven days. Each well was then reconstituted with 2.5 mL of Brugge medium, covered, and placed into incubation as previously outlined. On days 7, 14, and 21 post-reconstitution, paired 250 µL samples were obtained from each well for DNA extraction and qPCR amplification, and 100 µL from each well was plated onto TSA with 5% sheep blood (Figure 1c). Colony growth was monitored every four to seven days, and visible colonies were removed via sterile inoculating loop into Buffer AL (Qiagen, Germantown, MD, USA, cat.# 19075) for DNA extraction, PCR interrogation, and DNA sequencing to confirm identification. 

### 2.7. Statistical Analysis

Data analysis of bacterial concentration (GE) based on qPCR Ct value, including mean and standard error from paired sample evaluation, was performed using Microsoft Excel 2019 (version 16). Paired-sample *t*-tests were used to determine whether there was a significant change (*p* < 0.05) in bacterial DNA concentration over inoculation time within each fluid, and unpaired *t*-tests were used to assess for differences between fluid types (GraphPad Prism 9, San Diego, CA, USA). Fold change ratios were also calculated within fluid inoculates over time and are reported as the difference between the measured and original inoculate concentration divided by the original inoculate concentration. 

## 3. Results

### 3.1. Pre-Inoculation Evaluation of the Fluid Matrices for Bh SA2 DNA and Bacterial Growth

Pre-*Bh* SA2 inoculation screening for the presence of *Bartonella* species DNA by qPCR amplification targeting *Bartonella* 16S-23S ITS region was negative for all fluid matrices. No Colony growth was detected one week after feline urine, cow’s milk, physiologic saline solution, or sterile Brugge medium were inoculated onto blood agar plates; however, feline blood and serum grew contaminant bacteria when plated onto blood agar. These colonies were qPCR-negative for *Bartonella* DNA. Contaminant bacterial growth was attributed to a lack of sterile technique utilized by the commercial distributor during blood collection or subsequent sample pooling. The feline serum was IFA reactive for antibodies to *B. henselae* (1:512), *B. vinsonii* subsp. *berkhoffii* (1:512), and *B. koehlerae* (1:256), suggesting prior exposure to a *Bartonella* spp. 

### 3.2. Bartonella Henselae SA2 Viability and Stability in Six Fluid Matrices

Bacterial colonies were observed from all fluid inoculates at all culture time points, except for *B. henselae* SA2 growing in urine, where colony formation was observed on the 0 h and 24 h blood agar cultures (Table 1).

These results documented bacterial viability within all fluid matrices, indicating that the *Bh* SA2 organisms remained viable at the time of desiccation for all fluids other than urine. Bacterial isolate identity was confirmed by PCR amplification and DNA sequencing for all inoculated matrices (Eton Bioscience, Research Triangle Park, NC, USA). Despite the growth of contaminant bacteria from blood and serum at the time of inoculation, small colonies, confirmed as *Bh* SA2 by DNA sequencing, were visible on the respective blood agar plates from these two matrices at all time points. Blood agar plate colonies were never visualized from the negative control Brugge medium flask, co-incubated alongside the *Bh* SA2 inoculated matrices. 

To assess the growth or stability of *Bh* SA2 in each inoculated fluid, bacterial concentration was measured by DNA amplification using qPCR, as described in Section 2.4. The results are depicted in Figure 2. 

Variation was anticipated based on whether the fluid supported bacterial growth and stability or caused damage resulting in bacterial death. Although not a statistically significant decrease (*p* = 0.198), serum bacterial DNA concentration dropped 9.8-fold from time 0 h to 24 h and did not subsequently rebound. Unpaired *t*-tests (Table 2), however, demonstrated a statistically higher bacterial concentration between 24 h and 7 d of incubation in each of the other matrices compared to serum, with no difference between any of the fluids when GE were compared at time 0 h. There was no amplification of *Bh* SA2 DNA from the negative control. 

### 3.3. Bartonella henselae SA2 Viability and Stability in Fluid Matrices after Culture Enrichment with Brugge Media

Colonies, confirmed as *Bh* SA2 by DNA sequencing (Eton Bioscience, Research Triangle Park, NC, USA), developed from all original fluid cultures when placed into Brugge medium for bacterial culture enrichment (Table 3).

From the inoculated blood matrix, colonies developed from all time points following Brugge media enrichment, with the exception of *Bh* SA2 incubated in feline blood for 7 days, which produced colonies only at the 7-day evaluation. Milk and the Brugge positive culture control had identical colony growth patterns: both had colony development from all initial incubation times when evaluated at 7 and 14 days following inoculation into Brugge media, while colonies developed at 21 days only after *Bh* SA2 had incubated in the respective fluids for 96 h and 7 days. *Bh* SA2 incubated in serum and saline solution had variable colony formation following incubation in Brugge medium. For serum, colony formation was observed in all but the 0 h culture after 7 days of Brugge enrichment, whereas only the 96 h and 24 h cultures developed colonies following Brugge enrichment for 14 and 21 days, respectively. *Bh* SA2 incubated in saline for 0 h through 48 h developed colonies after 7 and 14 days of Brugge support, while colonies were observed from the 7-day saline inoculate following 7 and 21 days of Brugge enrichment. In contrast, urine cultures only yielded colony formation directly following inoculation (0 h). No colonies were visualized from the *Bh* SA2 un-inoculated (negative control) Brugge medium flask.

Bacterial concentration (GE) was measured in each of the Brugge-enriched fluid inoculates to assess for growth, decline, or stability. The results are depicted in Figure 3. 

*Bh* SA2 inoculated blood displayed the largest range of amplified DNA concentration after being sub-cultured in Brugge medium, with a positive trend in the 0 h, 96 h, and 7 d inoculates evaluated after 7–21 days of Brugge enrichment (Figure 3a). DNA concentration of *Bh* SA2 incubated in feline blood for 24–48 h remained relatively stable across all timed measurements. When averaged over the three weeks of Brugge enrichment, the blood culture *Bh* SA2 concentrations incubated for 24 h through 96 h were significantly higher than the measurable DNA from the 7 d *Bh* SA2 inoculate over the same time period (24 h *p* = 0.0167, 48 h *p* = 0.0002, 96 h *p* = 0.0335) (Figure 3a). *Bh* SA2 incubated in serum with Brugge enrichment displayed the lowest measured bacterial concentration, with a decline in detectable DNA between 0 h and 7 d, similar to what was observed in Figure 2 (Figure 3b). Average bacterial concentration in serum across the 21-day Brugge supplementation was higher for the 0 h inoculate compared to each other timed inoculate using a paired-samples *t*-test, although not significant by the 24 h measurement (24 h *p* = 0.466, 48 h *p* = 0.0368, 96 h *p* = 0.0300, 7 d *p* = 0.0231) (Figure 3b). Aside from a positive trend seen in the 0 h urine culture between 7 and 14 days of Brugge enrichment in Figure 3c, detectable DNA remained low and without variation across the timed interrogations. When *Bh* SA2 was cultured in milk, there was no discernible pattern in bacterial concentration between the length of incubation and Brugge enrichment (Figure 3d). Average bacterial concentrations across the 21 days of Brugge enrichment illuminated that *Bh* SA2 from the 0 h milk inoculation was higher than the subsequent timed measurements, but only significantly higher than the 96 h milk incubation time point (*p* = 0.0208). When *Bh* SA2 was inoculated in saline with Brugge enrichment, DNA detection was low, dropping on average by 0.76-fold across the 21-day period (Figure 3e). Except for the 0 h inoculation, the positive Brugge control matrix evaluated at the 7, 14-, and 21-day time points (Figure 3f) displayed a slight positive trend in growth between the 7-day and 21-day measurements, with the highest concentrations reached following incubation of *Bh* SA2 in Brugge medium for 24–48 h, followed by a sharp decrease in DNA concentration by 96 h. Statistically, both the averaged 24 h inoculate and the 48 h inoculate had higher *Bh* SA2 DNA concentrations than either the 96 h or the 7-day inoculate by paired-samples *t*-testing (*p* = 0.0329 (96 h) and *p* = 0.0482 (7 d) for the comparison to the 24 h culture and *p* = 0.0014 (96 h) and *p* = 0.0036 (7 d) when compared to GE averaged across the 48 h inoculate). No amplification of *Bh* SA2 DNA was observed from the negative Brugge medium control. 

### 3.4. Bh SA2 Viability and Stability after Desiccation of Inoculated Fluid Matrices and Reconstitution with Brugge Growth Medium 

Following air-desiccation and incubation in Brugge medium, colony growth was obtained from all reconstituted fluid matrices (Table 4).

As paired 6-well plates were utilized for the desiccation component of the experiment, a blood agar culture was established from each paired well. Compared to the other reconstituted matrices, colonies grew most consistently from blood and serum cultures plated following desiccation and reconstitution with Brugge medium, regardless of the inoculate incubation duration time. For milk, saline, and the positive Brugge control, no colonies grew from the 0 h desiccated cultures, but growth was obtained from these three matrices following Brugge reconstitution from the longer incubation periods (Table 4). For urine, only a single timepoint post-desiccation and reconstitution resulted in colony growth: the 24 h inoculate incubated in Brugge medium for 14 days. From a representative sample group of positive blood agar cultures (*n* = 23), DNA was extracted, amplified as previously described, and submitted to Eton Bioscience (Research Triangle Park, NC) for genomic sequencing. Based on qPCR melting curves and the DNA sequences obtained, colonies growing post-desiccation and reconstitution were *Bh* SA2, which was the species and strain used as the inoculum for the experiment. There were no colonies observed from the negative control. 

Amplification of *Bh* SA2 DNA by 16S-23S qPCR was attempted for all inoculates at three time points: 7, 14, and 21 days following Brugge medium reconstitution (Figure 4). 

For the desiccated feline blood culture, there was a trend towards increased *Bh* SA2 DNA following Brugge reconstitution between the 7- and 14-day measurements for most of the original culture testing time points; however, only the 0 h culture time point approached statistical significance (*p* = 0.0509). Compared to the concentration of *Bh* SA2 measured prior to desiccation, the 0 h and 24 h inoculates were higher, although not significantly (*p* = 0.0720 and 0.2006, respectively). However, the average concentration of the 0 h and 24 h inoculates following desiccation and reconstitution compared to their pre-desiccation concentration was significantly higher (*p* = 0.0061) (Figure 4a). The 14-day blood incubation period yielded the highest DNA level in the 0 h inoculate, which was significantly higher than that attained at 14 days from the 48 h–7 d incubation times (*p* = 0.0480). For the other fluid matrices, DNA concentrations remained lower than those detected in the pre-desiccation inoculates. *Bh* SA2 inoculated serum had the lowest residual DNA concentration among the six matrices, with no discernibly consistent DNA amplification pattern (Figure 4b). Average *Bh* SA2 DNA concentrations measured across all the time points were not significantly different for the serum or milk inoculates (Figure 4d). There was a significant increase in average GE for the 7 d saline inoculate between the 7- and 21-day interrogations compared to the other timed incubations in that matrix (*p* = 0.0254) (Figure 4e). For the *Bh* SA2 desiccated urine, the highest DNA concentration was measured after 14 days of Brugge enrichment for the 0 h urine inoculate, followed by the 21-day measurement for the 7 d desiccated urine culture. The average *Bh* SA2 DNA in both the 0 h and 7 d urine cultures exceeded that obtained from the 24–96 h cultures (*p* = 0.0020 and 0.0022, respectively) (Figure 4c). Of interest, the desiccated 24 h urine inoculate after 14 days of Brugge enrichment was the only time point at which urine colony growth was visualized (Table 4). This growth occurred from only one of the two desiccated inoculates, and when compared against the genome equivalents obtained from each of these wells, correlated with the higher DNA concentration (9.54 × 10^5^ vs. 4.38 × 10^5^ GE/µL). In general, the GE graphs for milk and saline had similar patterns. Both had the highest level of GE obtained the longer the matrix-inoculated bacteria were incubated prior to desiccation (7 d inoculate). As well, although small, these inoculates trended towards an increase in GE over the period of the 21-day Brugge enrichment time. For milk, there was a percent variance increase of 7.73% between the 7- and 21-day incubation periods, compared to a change of 2.47% for saline (Figure 4d,e). The desiccated positive control (Brugge) had an increase in amplifiable DNA in the 96 h and 7-day inoculates between the 7- and 21-day reconstitution evaluations (*p* = 0.0370 and 0.0059, respectively) (Figure 4f). The negative control resulted in no amplification of *Bh* SA2 DNA. 

## 4. Discussion

In this study, *B. henselae* strain San Antonio 2 was successfully cultured in cow’s milk, sterile physiological saline solution, and three body fluids derived from cats, i.e., blood, serum, and urine. Following desiccation by air-drying and reconstitution in Brugge medium for culture enrichment, viable *Bh* SA2 grew from each fluid matrix. In the context of environmental stability, our results support bacterial persistence in various physiological fluids, as well as the ability to survive desiccation by air-drying. Following inoculation of *Bh* SA2 into the five fluid matrices and the Brugge positive control flask, *Bh* SA2 DNA was amplified and sequenced from all six liquid cultures at time points spanning 0 h to 7 days (Figure 2). During this period, there was no denaturation of bacterial DNA in five of the six fluid matrices. However, when cultured in cat serum, amplifiable *Bh* SA2 DNA remained below the inoculated concentration following 24 h of incubation, possibly related to the presence of anti-*Bartonella* species antibodies in the combined serum samples utilized, as the comparable times measured across the other test fluids did not result in a similar pattern. In contrast, all other inoculated matrices resulted in significantly higher *Bh* SA2 concentrations during the 7-day incubation period (Table 2). Contrary to our hypothesis, *Bh* SA2 colonies were isolated from inoculated milk, saline solution, and serum, as well as from blood, at all time points between inoculation (0 h) and 7 days of incubation, and from urine at 0 h and 24 h, underscoring bacterial viability within these matrices (Table 1). Although some loss of bacterial viability occurs when transitioning from a liquid environment (growth medium) to a solid medium, we did not confirm viability by isolation in feline urine past 24 h; however, since *Bh* SA2 colonies were also obtained from the 24 h desiccated urine culture after 14 days of Brugge enrichment, it seems likely that these bacteria are also more viable in urine than has been previously reported [80]. Future research could include an evaluation of RNA (through reverse transcription qPCR using 16 s rRNA as a target) to help further determine the extent and duration of *B. henselae* viability in urine. 

Bacterial viability in each fluid matrix was also assessed by sub-culture using Brugge medium (specifically used to support and enrich *Bartonella* growth) [79] followed by attempted blood agar plate isolation. Colonies of *B. henselae* SA2 were obtained from Brugge sub-cultured blood, milk, saline solution, and serum. *Bartonella* colony formation was similar between these inoculated matrices and, to a lesser extent, saline when compared to colony isolation following Brugge medium enrichment. *Bh* SA2 inoculated into blood resulted in a higher bacterial DNA concentration across the 21-day Brugge supplementation period for the 24 h–96 h blood cultures compared to the 7 d blood culture (Figure 3), where less colony growth was visualized at this time point (Table 3). It is anticipated that *Bh* SA2 cultured into feline blood for 7 days may have experienced nutritional deficits, resulting in this observation. For milk, the 0 h culture tested at 7, 14, and 21 days had significantly higher bacterial concentrations than the 96 h culture, but in contrast to blood, the 96 h milk culture produced colonies at all measured time points, emphasizing a disconnect between amplifiable DNA and the development of viable colonies. Serum inoculated with *Bh* SA2 resulted in colony formation in all but the 0 h culture after 7 days of Brugge enrichment (Table 3). Selective pressure from anti-*Bartonella* antibodies may have afforded those bacteria surviving in serum for longer periods a temporary growth advantage. Evaluation of differential gene expression in *B. henselae* cultured in the presence and absence of antibodies represents another interesting area for future endeavors, with the possibility of identifying diagnostic or therapeutic targets. With the exception of *Bh* SA2-inoculated blood, increased variability in colony formation was observed across cultured matrices enriched with Brugge medium for 21 days. This finding may be attributed to bacterial death secondary to nutrient limitations, as the medium was not refreshed throughout the experimental time span. Similar to the matrix fluid cultures, *Bh* SA2 DNA was amplified from all Brugge-enriched cultures (Figure 3), despite the lack of bacterial isolation for some enriched fractions, denoting DNA stability among the Brugge-supplemented matrices. These in vitro findings reiterate the comparison value of DNA detection via PCR when considering diagnostic relevance. From serum, colonies were visualized from the 96 h culture after 14 days of Brugge culture enrichment (Table 3), yet DNA concentration measured at this point was 4.0 × 10^4^ genome equivalents per µL, one of the lowest measured concentrations among the serum series (Figure 3b), a scenario that was replicated in comparing colony growth to extracted bacterial DNA concentration in other fluids. Colony formation was not obtained when *Bh* SA2 incubated in feline urine was inoculated into Brugge medium, with the exception of colony growth from the 0 h inoculate (Table 3), despite 21 days of culture enrichment. Our results suggest that either feline urine does not support the presence of viable *Bh* SA2 over prolonged periods, or that the method of analysis used in this study was inadequate to assess viability in urine. 

Unexpectedly, *Bh* SA2 was viable after bacterial cultures in blood, serum, urine, milk, saline solution, and the Brugge(+) control were allowed to air-desiccate. When Brugge medium was used to recover dried cultures, colonies were observed at variable testing time points from almost all of the desiccated culture fluids (Table 4), with blood being the most supportive of bacterial recovery. For the urine culture, colony development following desiccation occurred from only the 24 h inoculation. As direct culture onto blood agar from inoculated urine over time is more suggestive of the inability of urine to support *Bh* SA2 (Table 1), it is anticipated that colony development, in this case, was due to the short length of time the *Bartonella* inoculate spent in that matrix, combined with the additional support of Brugge medium for culture enrichment. It is interesting, however, that live colonies were attained from urine following desiccation, as *Bartonella henselae*, unlike *Bartonella quintana*, which was deemed infectious through cutaneous inoculation of patient urine, has not been determined to be transmitted through the urine of infected individuals [80]. Assessment of *Bartonella*’s ability to persist in urine, along with mechanisms of survival, outlines another area for future research. For saline and milk, colony development following desiccation was successful the longer the original inoculum was exposed to those matrices (Table 4). Here, it is possible that bacteria were able to acclimate to these environments long enough to allow for the development of protective mechanisms. Future research directed at determining the mechanism(s) of desiccation resistance in this and potentially other *Bartonella* species is warranted. As previously observed in the other experimental conditions in this study, *Bartonella* DNA was amplified from all post-desiccation/reconstitution time points (Figure 4), again pointing to the stability of bacterial DNA despite air-desiccation. Interestingly, for the 0 h and 24 h blood cultures, there was an increase in bacterial genome equivalents between the 7- and 14-day interrogations, with averaged genome equivalents between these times reaching statistical significance, yet again, colony development on blood agar did not strongly correlate with the measured DNA concentrations (Table 4). Similarly, amplifiable DNA from *Bh* SA2 incubated in feline urine for 0 h and 7 d prior to desiccation and reconstitution was higher than at the other interrogation times, yet colonies developed from only the 24 h inoculate. In saline, the average DNA concentration was greatest when *Bh* SA2 was incubated for a period of 7 days prior to desiccation and reconstitution, and here, the 7 d inoculate more consistently resulted in colony development compared to the other interrogation times. Following desiccation and reconstitution, the 96 h and 7 d Brugge control inoculates had significant increases in DNA between the 7-day and 21-day measurements (Figure 4), more indicative of bacterial growth, which better correlated with the colony observations obtained at these time points (Table 4).

Traditionally, and for diagnostic purposes, blood has been the primary target sample type for the detection of *Bartonella* species in reservoir hosts and incidentally infected patients. Although few reports describe the detection of *Bartonella* DNA from other body fluids such as serum [56,81], cerebrospinal fluid (CSF) [82,83], lymph node aspirates [84], aqueous humor [85], urine [86], and saliva [64,65,86,87], to our knowledge, a systematic assessment of the viability of this or other *Bartonella* species in these diagnostic patient fluid specimens has not been undertaken. Historically, case reports describing people acquiring CSD from rose thorn injuries [88], as well as cat or dog salivary transmission from bites and scratches [89,90,91], point to this organism’s ability to tolerate a wide range of environmental conditions. The fact that *B. henselae* can be revived from several fluid matrices following desiccation not only disproved our hypothesis but supports the hypothesis of previously unrecognized infectious sources and alternative modes of transmission. In the context of occupational risk, these findings are of particular concern for veterinary medical professionals and other individuals or professions with extensive animal exposure [92]. Whether exposure to desiccated bacteria presents a medical risk for humans and other animals remains unknown; however, our findings warrant the need for consideration of non-vectorial modes of transmission in future laboratory, clinical, and epidemiological studies. *B. henselae* DNA, along with DNA from other human pathogenic *Bartonella* species, has been amplified from dust mites (*Dermatophagoides farinae*) and their feces [93,94], and *Bartonella koehlerae*, a cat-associated species related to blood culture-negative endocarditis in humans [95], has been associated with respiratory symptoms [96]; therefore, a potential role for inhalation of dead or viable *Bartonella* spp. in asthma or other respiratory conditions should be investigated. Although dust mites and their excrement have not been evaluated for potential vector capacity or as an environmental source of *Bartonella* infection, if *B. henselae* can survive desiccation in a natural setting, this may represent an unexplored repository of infectious material. Importantly, viability following desiccation could provide the bacterium with extended time for environmental transmission to a mammalian reservoir host, insect vector, or incidental host, accentuating the need to further assess the environmental transmissibility of this bacterium.

The ability of *B. henselae* to survive as planktonic microorganisms in fluids other than blood has identified the possible existence of an unexplored environmental niche for this bacterium in nature. Multiple worldwide studies have elucidated coinfections of *Toxoplasma gondii* and *B. henselae* in domestic and wild felids, with *B. henselae* commonly being the pathogen with a higher prevalence [47,97,98,99]. Like *T*. *gondii*, following the shedding of *B. henselae* into terrestrial and aquatic environments, either via fleas, flea feces from infected cats, or infected bodily fluids from cats or other terrestrial animals [73], the bacteria might be capable of surviving in regional watersheds or ocean water, as evidenced by bacterial viability within physiologic saline. It remains unknown whether ingestion of one or more intermediary aquatic species, vector transmission, or inoculation of wounds with contaminated water might be associated with the acquisition of *B. henselae* infection by cetaceans [66,67,70], and this may be an exciting avenue of future study. Also, the viability of *B. henselae* in feline blood and serum and bovine milk for up to seven days could represent sources of environmental contamination. 

In this study, we examined the ability of *Bh* SA2 to remain viable in several fluid matrices for up to seven days, independent of its flea vector or primary mammalian reservoir host, and to survive desiccation in physiological fluids, when reconstituted using an enrichment culture medium. Comparison of agar plate isolation with PCR amplification of *B. henselae* DNA illustrated a lack of correlation between these two methodologies, which has been noted in previous studies [42,46,62]. Despite these important preliminary findings, there were limitations associated with this study. First, only a single strain of *Bartonella henselae* (San Antonio 2 strain type) was used for all experiments. The inoculum bacteria were accustomed to growing in Brugge medium under laboratory conditions, which may have influenced its ability to survive in other chosen fluid matrices and may have impacted bacterial desiccation tolerance [100,101]. For comparison to the results from this study, future studies should be performed using multiple strains of *B. henselae* as well as different species of *Bartonella*, including recently isolated wild-type bacteria. This could provide not only confirmation for our observed results, but also potentially delineate the differential ability of other *Bartonella* strains and species in response to the experimental conditions. Replicates were collected in duplicate, most likely negatively impacting statistical power for comparison of results. Although hypothesis-based, this study was observational—we did not ascertain growth kinetic curves based on colony formation unit (cfu) enumeration due in part to the difficulty of assessing singular colonies after plating from liquid media to blood agar, and due to the slow rate of growth of this organism impacting colony visualization [42,46,61]. Though mechanisms that support the viability of *Bh* SA2 in the various fluid matrices and allow it to survive desiccation are suspected to be related to biofilm production, this was not investigated and remains the topic of future investigation [102]. To assess bacterial persistence and viability, a large inoculation concentration was utilized, much higher than would be present during natural infection of a flea or feline reservoir host. To approximate more closely what might happen in a natural setting, lower inoculum concentrations should be evaluated. Bacterial contamination in the commercially purchased feline blood and serum was present, and although this did not prevent the growth of *Bh* SA2, we cannot rule out a synergistic or antagonistic effect on bacterial viability. Potentially, heat inactivation of blood and serum could be used for future experiments to decrease the risk of bacterial contaminants. Lastly, these experiments were completed in a highly controlled in vitro laboratory setting that does not replicate numerous factors that would likely impact the potential for bacterial environmental contamination and stability. Despite these limitations, the information gained from these experiments provides a starting point for future endeavors to clarify the stability and viability of *B. henselae* outside of a mammalian host or arthropod vector, in terrestrial and aquatic environments, including the assessment of virulence genes and biofilm formation following inoculation into liquid matrices and after desiccation.

## 5. Conclusions

As biomedical research publications continue to increase the collective understanding of this insidious pathogen, *B. henselae* is proving to have previously unrecognized environmental survival capabilities [103,104]. There remains a substantial need to better understand the breadth and depth of *Bartonella* species’ zoonotic disease ecology as it pertains to animal and human health in diverse terrestrial and marine environments. Additionally, still substantially unexplored is the potential role that *Bartonella* species may play in diseases affecting wildlife, or its impact on biodiversity [105,106,107,108]. Research documenting the presence of *B. henselae* DNA in cetacean blood and tissue may represent only the tip of the proverbial iceberg for marine environments. Also, knowledge of its environmental stability should add *Bartonella* to the list of pathogens that need to be investigated in association with sylvatic disease outbreaks. The impact of global climate change on vector range, in combination with urbanization and loss of wildlife habitat, may equate to people more routinely coming into contact with *Bartonella* species [109,110,111]. If indeed this pathogen can switch between an indirect means of infection through its vector and a direct means of infection, then researchers need to remain vigilant to its potential to cause disease in animals and human patients in conjunction with a new epidemiologic transmission paradigm. In considering the potential for *Bartonella henselae* to exhibit a higher level of environmental stability, as well as the possibility of the bacteria remaining viable outside of its mammalian host or insect vector for prolonged periods of time, it is important to utilize a One Health approach that addresses the comparative aspects of human and animal infections, as well as impacts of various vectors and the environment, to achieve a better understanding of *B. henselae* transmission in nature.

## 6. Patents

In conjunction with Dr. S. Sontakke and North Carolina State University, E. B. Breitschwerdt holds US Patent No. 7,115,385 Media and Methods for Cultivation of Microorganisms, which was issued on 3 October 2006. 

## Figures and Tables

**Figure 1 pathogens-12-00950-f001:**
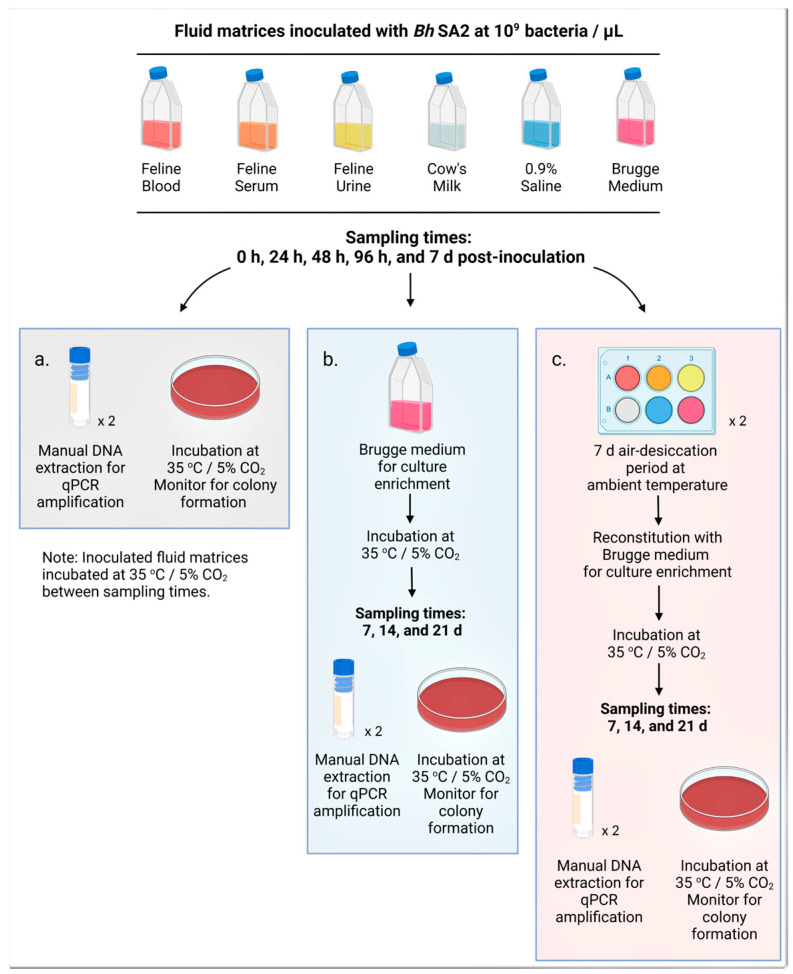
Methods overview for studying viability and desiccation resistance of *Bartonella henselae* in various fluid matrices. First, 10 mL each of feline whole blood, serum, and urine; cow’s milk; and physiologic saline were inoculated with *Bartonella henselae* strain San Antonio 2 to reach a concentration of 10^9^ bacteria per µL. Samples were obtained from each of the inoculated fluids at time 0 h, 24 h, 48 h, 96 h, and 7 days as follows: (**a**) Paired 250 µL aliquots were placed into 1.8 mL cryovials for DNA extraction and qPCR amplification, and 100 µL plated onto Trypticase Soy Agar (TSA) with 5% sheep blood, incubated, and monitored for colony development. (**b**) A total of 100 µL from each fluid inoculum was incubated in 5 mL of Brugge medium for bacterial culture enrichment. After 7, 14, and 21 days of incubation, samples were obtained for DNA extraction and blood agar plate inoculation as described in (**a**). (**c**) Paired 250 µL aliquots were placed into individual wells of paired 6-well plates, desiccated overnight in a biosafety laboratory level 3 (BSL-3) vented biosecurity cabinet, then fitted with lids and transferred to an enclosed benchtop container at ambient temperature for the remainder of 7 days. On day 7, the desiccated material was reconstituted using 2.5 mL of Brugge medium, and the 6-well plates were placed under incubation. After 7, 14, and 21 days of incubation, each well of the paired 6-well plates was sampled as outlined in (**a**). The negative control fluid, uninoculated Brugge medium, was sampled and stored alongside the test fluids. Note: Samples for manual DNA extraction were stored at −20 °C pending extraction, and all incubated samples were kept in a dedicated incubator at 35 °C with 5% CO_2_. Figure created in BioRender.com (https://app.biorender.com/ accessed on 29 August 2022).

**Figure 2 pathogens-12-00950-f002:**
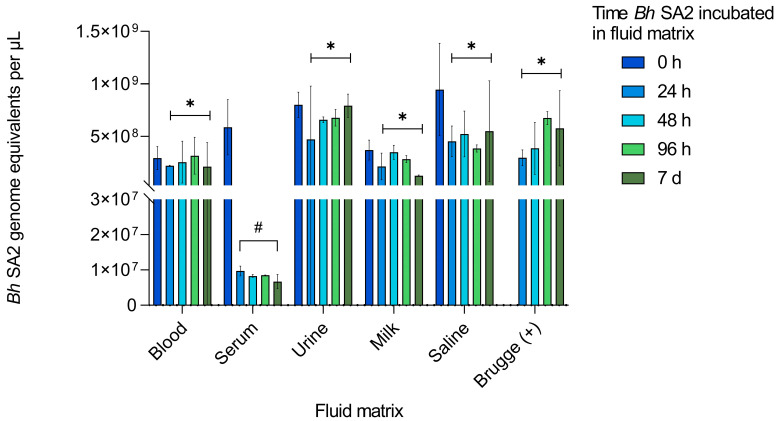
Concentration of *Bartonella henselae* SA2 DNA identified following incubation in feline blood, serum, or urine; cow’s milk, physiological saline, and Brugge enrichment medium over time. Paired measurements were averaged and are depicted as GE per µL with standard error of that fluid’s mean concentration. The concentration of *Bh* SA2 in inoculated feline serum was lower between 24 h and 7 d (#) than that measured in the other fluids (* *p* ≤ 0.001), but no significant difference was found for the 0 h GE across fluids. Note: The Brugge positive control bacterial GE at 0 h was not determined due to sample loss.

**Figure 3 pathogens-12-00950-f003:**
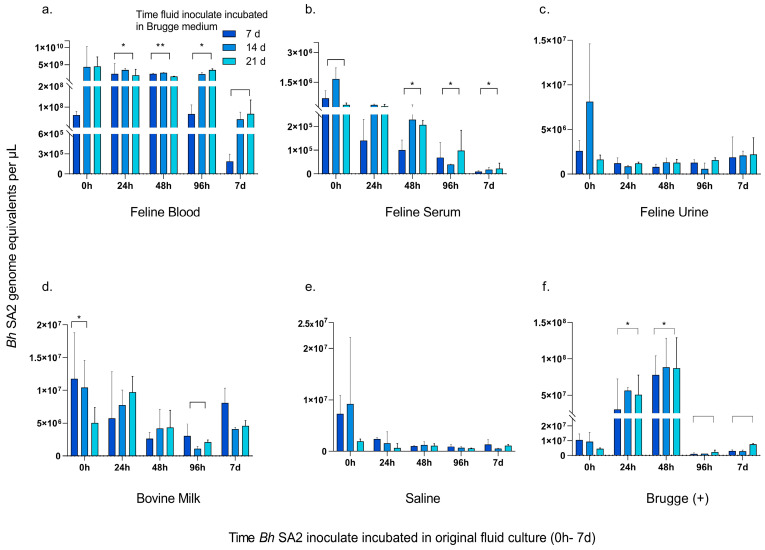
Concentration of *Bartonella henselae* SA2 DNA amplified after incubation of inoculated fluid matrices with Brugge medium for culture enrichment. Note the y-axis scale differs between fluid matrices. Concentration is plotted as bacterial GE per µL for each fluid inoculum at 7, 14, and 21 days following incubation in Brugge medium for culture enrichment. (**a**) The average concentration across the 7–21-day measurements for *Bh* SA2 incubated in feline blood prior to Brugge supplementation for 24 h*, 48 h**, and 96 h* was higher than the concentration achieved from *Bh* SA2 incubated in blood for 7 d prior to Brugge enrichment (* *p* < 0.05, ** *p* < 0.001). (**b**) There was a decline in *Bh* SA2 concentration (* *p* < 0.05) when incubation in feline serum was averaged across the 7–21-day period. (**c**) *Bh* SA2 concentrations attained in feline urine over time. (**d**) For the bovine milk inoculate, the average time 0 h *Bh* SA2 concentration over the 7–21-day Brugge enrichment period was significantly higher than the 96 h *Bh* SA2 incubated culture (* *p* < 0.05). (**e**) *Bh* SA2 concentration in physiologic saline following enrichment with Brugge media. (**f**) For the positive control, incubation periods of 24 and 48 h averaged significantly higher concentrations across the Brugge enrichment period compared to the 96 h inoculate and 7 d inoculates, respectively (* *p* < 0.05).

**Figure 4 pathogens-12-00950-f004:**
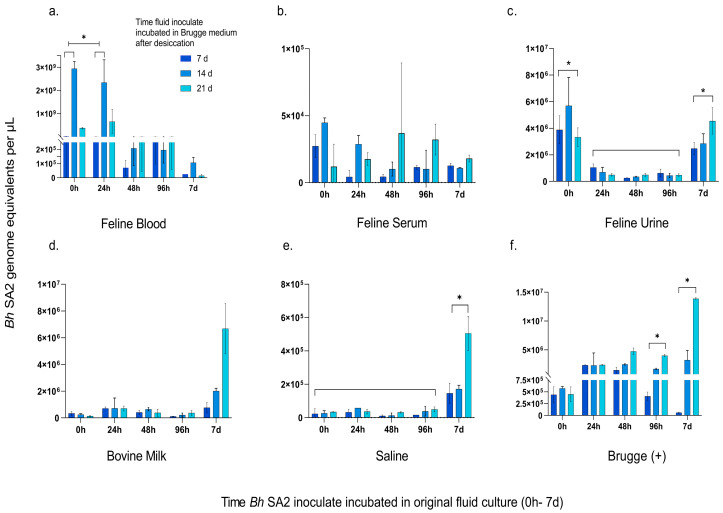
Concentration of *Bartonella henselae* SA2 recovered from reconstituted fluid matrices at 7, 14, and 21 days. Note the y-axis scale differs between fluid matrices. Bacterial concentration (GE) from each reconstituted fluid culture is plotted against the time that *Bh* SA2 incubated in each test fluid, with measurements 7, 14, and 21 days after incubation in Brugge medium. (**a**) There was no difference between the 0 h and 24 h feline blood cultures between 7 and 14 days of Brugge incubation when evaluated individually; however, the averaged 0 h–24 h concentrations increased between 7 and 14 days following incubation in Brugge medium (* *p* < 0.05). The 0 h inoculate 14 d measurement was also significantly greater than the same measurement in the 48 h–7 d fluid samples (*p* < 0.05). (**b**) Concentration of *Bh* SA2 recovered from the reconstituted serum inoculate. (**c**) *Bh* SA2 recovered from the reconstituted urine inoculate had an average concentration in the 0 h culture and 7 d culture over the 7–21-day period that was significantly higher than the 24 h–96 h cultures (denoted with brackets) (* *p* < 0.05). (**d**) Concentration of *Bh* SA2 recovered from reconstituted milk inoculate. (**e**) *Bh* SA2 concentration recovered from reconstituted saline was significantly increased for the 7 d inoculate over the 21-day Brugge enrichment period compared to the other inoculate times (denoted with brackets) (* *p* < 0.05). (**f**) *Bh* SA2 concentration from the reconstituted Brugge medium (+) control incubated for 96 h and 7 days was significantly increased between the 7- and 21-day measurements (* *p* < 0.05).

**Table 1 pathogens-12-00950-t001:** Growth of *Bartonella henselae* colonies on blood agar following incubation in feline blood, serum, or urine; cow’s milk; physiologic saline; or Brugge medium for up to 7 days. Colonies identified as *Bh* SA2 through DNA sequencing are denoted as (+). No colonies were cultured from the feline urine after 24 h of bacterial incubation, and no colonies developed from the negative control. Blood agar plates were incubated for 4 weeks to allow sufficient time for bacterial growth.

Time *Bh* SA2 Incubated in Fluid Matrix					
Inoculated Fluid Matrix	0 h	24 h	48 h	96 h	7 d
Blood	+	+	+	+	+
Serum	+	+	+	+	+
Urine	+	+			
Milk	+	+	+	+	+
Saline	+	+	+	+	+
Brugge	+	+	+	+	+

**Table 2 pathogens-12-00950-t002:** Unpaired *t*-test results (*p*-value) for comparison of measurable *Bartonella henselae* SA2 DNA in inoculated feline serum compared to the other fluid matrices following 24 h to 7 days of incubation in each fluid. All other fluid matrices had significantly higher bacterial GE measurable over 24 h–7 d following bacterial inoculation compared to serum. No significant difference existed in GE measured at time 0 h between any fluid matrix.

Inoculated Fluid Matrix	*p*-Value
Blood	0.001
Urine	0.0001
Milk	0.0004
Saline	0.0005
Brugge	0.0007

**Table 3 pathogens-12-00950-t003:** *Bartonella henselae* SA2 colony formation obtained from each fluid matrix after sub-culture and incubation in Brugge enrichment medium for up to 21 days. Positive cultures, identified as *Bh* SA2 by DNA sequencing, are denoted with a (+).

Time *Bh* SA2 Incubated in Original Fluid Matrix															
Inoculated Fluid Matrix	0 h	24 h	48 h	96 h	7 d	0 h	24 h	48 h	96 h	7 d	0 h	24 h	48 h	96 h	7 d
	Time Incubated in Brugge Enrichment Medium														
	7 days					14 days					21 days				
Blood	+	+	+	+	+	+	+	+	+		+	+	+	+	
Serum		+	+	+	+				+			+			
Urine	+														
Milk	+	+	+	+	+	+	+	+	+	+				+	+
Saline	+	+	+		+	+	+	+							+
Brugge	+	+	+	+	+	+	+	+	+	+				+	+

**Table 4 pathogens-12-00950-t004:** *Bartonella henselae* SA2 colony formation following air-desiccation in the various fluid matrices and reconstitution with Brugge medium. Positive colony growth denoted as (+). The presence of two (++) indicates colony development from both paired samples.

Time *Bh* SA2 Incubated in Original Fluid Matrix															
Inoculated Fluid Matrix	0 h	24 h	48 h	96 h	7 d	0 h	24 h	48 h	96 h	7 d	0 h	24 h	48 h	96 h	7 d
	7-day desiccation period at ambient temperature														
	7 days post-reconstitution					14 days post-reconstitution					21 days post-reconstitution				
Blood	+	++	+	+	+		++	+	+		+	+	++	+	++
Serum		++	++	++	++		+	+			+		++		+
Urine							+								
Milk			+	+	++			+				+	+	+	++
Saline				+	++							+			+
Brugge				++	++							+	+		+

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
