# Peer review of "Viability and Desiccation Resistance of *Bartonella henselae* in Biological and Non-Biological Fluids: Evidence for Pathogen Environmental Stability"

_pathogens, 2023, doi:10.3390/pathogens12070950_

Round 1
Reviewer 1 Report
There is no serious concerns about the manuscript. However, If you can challenge more Bartonella henselae strains or other Bartonella species in the experimental conditions which you had prepared for the present study, this conclusions should be more confirmed. I recommend you to included this advices, if you try to do so for future. Throughout the whole, I guess that the evidences obtained from this study are likely to be useful for preventing accidental B. henselae infection.
Author Response
The authors wish to thank the reviewer for their thoughtful comments on the manuscript. To address the advice to confirm our results utilizing other Bartonella strains and species, the following has been added to the discussion:
For comparison to the results from this study, future studies should be performed using multiple strains of B. henselae as well as different species of Bartonella, including recently isolated wild-type bacteria. This could provide not only confirmation for our observed results, but also potentially delineate differential ability of other Bartonella strains and species in response to the experimental conditions.
We are planning to evaluate other Bartonellas that act as human pathogens in attempts to determine environmental stability.
Thank you very much for your review of our manuscript.
Reviewer 2 Report
Dear author,
The manuscript title is “Viability and Desiccation Resistance of Bartonella henselae inBiological and Non-biological Fluids: Evidence for Pathogen-EnvironmentalStability” and it aims to assess the likely ability of B. henselae to survive in different fluid matrices.
The topic falls within the aims and scope of the journal. It is relevant in the field of zoonotic pathogens and One Health. It is very well supported by references, the results are clearly presented as well as their discussion.
Some particular suggestions/comments will be done here:
- Line 61 – You need to add a reference instead of “1”
- Lines 119 – 126 – I would transfer these lines to materials/methodology and results/conclusions as they are not background to be present in Introduction
- Please add the caption of the figures below/after the figure
- Line 130 – you will have to write somewhere in full TSA and BSL
- Line 134 – please write days instead of “d”
- Line256 – incubated?
- Along the manuscript, sometimes you have 7d others 7 day and even others 7 days. Please try to write the same way everytime it is possible to
- Line 574 – as CSF is written for the first time in the manuscript, write it full please
Author Response
The authors would like to thank this reviewer for their thorough evaluation of this manuscript. Each of the following concerns has been addressed, and written word changes or additions are highlighted in yellow in the manuscript. Line 61: The reference number 1 has been changed to [1] in the manuscript. Lines 119-126: The section on testing the hypotheses was moved to the beginning of the methods, and the segment on survival in the matrices and following desiccation was deleted as it was stated in the results section. The figure legends were relocated to after each figure. Line 130 was corrected for trypticase soy agar (TSA) and biosafety laboratory level 3 (BSL-3). Line 134 was corrected to days from "d". Line 256 was shortened to "incubated". Line 574: cerebrospinal fluid was added before (CSF). The use of d, day and days in desribing the 7 day cultures and incubation time was standardized for easier readability.
Thank you very much for your review and suggestions, we appreciate your attention in reading this manuscript.
Reviewer 3 Report
The manuscript entitled "Viability and Desiccation Resistance of Bartonella henselae in Biological and Non-biological Fluids: Evidence for Pathogen- Environmental Stability" is a very interesting study based on an original idea that could give very important information for Public Health if it is further explored.
Some minor points are as follows:
Serum testing for the presence of antibodies against Bartonella should be justified. Moreover, details on antibody detection conditions should be presented or a reference should be given. The cut-off value used is also important since pooled samples were tested and the concentration of antibodies of an exposed cat might have been lower due to the dilution caused by adding sera of negative cats.
Regarding figures the description usually follows the figure, and the size of graphs is rather small, therefore authors could try to put two graphs per row to make graphs more readable.
Author Response
The authors would like to thank this reviewer very much for their thorough assessment of our manuscript. In addressing your recommended edits, we added justification for the use of antibody screening in lines 165-167 in the methods section: As cats are the known reservoir of B. henselae, antibody screening was performed to assess for the presence of anti-Bartonella antibodies that could impact bacterial survival in serum. The method used for IFA was also added in this section to address the overall methods and potential limitations of the cut-off value due to the potential dilution of pooled serum from Bartonella negative and positive cats, as indicated in lines 174-176: End-point titers >/ 1:64 were considered positive in order to account for the potential of a dilution effect subsequent to the use of pooled serum samples from cats with unknown Bartonella exposure without overinterpretation.
We appreciate the reviewers recommendations for ease of graphic depiction readability. We reformatted the figures to have two graphs per row as opposed to three, however the individual graphic size did not change significantly. Our reasoning behind displaying these results with three graphs per row is to allow the reader to evaluate results from all of the feline fluid associated values in one row, compared to results from non-feline associated fluids on the second row. We are happy to change this, however, if the reviewer thinks that this presentation significantly impacts the readability due to size.
Thank you for your time and recommendations.